# Regulation of starvation-induced hyperactivity by insulin and glucagon signaling in adult *Drosophila*

**Yue Yu[1,2†], Rui Huang[1,2*†], Jie Ye[1,2], Vivian Zhang[3], Chao Wu[1,2], Guo Cheng[1,2], Junling Jia[1,2], Liming Wang[1,2*]**

[1]Life Sciences Institute, Zhejiang University, Hangzhou, China; [2]Innovation Center for Cell Signaling Network, Zhejiang University, Hangzhou, China; [3]Department of Molecular and Cell Biology, University of California, Berkeley, Berkeley, United States

**Abstract** Starvation induces sustained increase in locomotion, which facilitates food localization and acquisition and hence composes an important aspect of food-seeking behavior. We investigated how nutritional states modulated starvation-induced hyperactivity in adult *Drosophila*. The receptor of the adipokinetic hormone (AKHR), the insect analog of glucagon, was required for starvation-induced hyperactivity. AKHR was expressed in a small group of octopaminergic neurons in the brain. Silencing AKHR[+] neurons and blocking octopamine signaling in these neurons eliminated starvation-induced hyperactivity, whereas activation of these neurons accelerated the onset of hyperactivity upon starvation. Neither AKHR nor AKHR[+] neurons were involved in increased food consumption upon starvation, suggesting that starvation-induced hyperactivity and food consumption are independently regulated. Single cell analysis of AKHR[+] neurons identified the co-expression of *Drosophila* insulin-like receptor (dInR), which imposed suppressive effect on starvation-induced hyperactivity. Therefore, insulin and glucagon signaling exert opposite effects on starvation-induced hyperactivity via a common neural target in *Drosophila*.

**\*For correspondence:**
huangrui0716@sina.com (RH);
lmwang83@zju.edu.cn (LW)

[†]These authors contributed equally to this work

**Competing interests:** The authors declare that no competing interests exist.

## Introduction

Energy homeostasis is vital for survival, growth, and reproduction of animal species (*Gautron et al., 2015*). Energy deprivation drives a complex behavioral program to ensure adequate food intake (*Sternson et al., 2013*). As an adaptive response to starvation, food intake is modulated by nutrient and hormonal cues. In mammals, the hypothalamic arcuate nucleus, especially the neurons expressing agouti-related protein (AgRP)/neuropeptide Y (NPY), and those expressing pro-opiomelanocortin (POMC), sense and integrate numerous nutritional cues such as circulating glucose, insulin, leptin, and ghrelin and modulate food intake in response (*Belgardt et al., 2009*). In fruit flies, several hormonal signals modulate different aspects of food intake, these including *Drosophila* insulin-like peptides (DILPs), the two homologs of mammalian NPY, neuropeptide F (NPF) and short neuropeptide F (sNPF), and a handful of other neuropeptides such as allatostatin A (AstA), leucokinin, and hugin (*Pool and Scott, 2014*). It is generally accepted that these hormonal cues represent the nutritional status of animals and translate them into physiological and behavioral responses (*Pool and Scott, 2014*; *Sternson et al., 2013*).

The search of appropriate food sources (i.e. food seeking) is often the first step of food intake, followed by the actual ingestion of food (i.e. food consumption) (*Stephens et al., 2007*). Starvation enhances food-seeking behavior in two parallel ways. On the one hand, starvation modulates the perception of food associated sensory cues, increasing the likelihood to target desirable food

**eLife digest** Animals can be thought of as tightly controlled eating machines. An animal's brain senses if it is hungry via signals from the nervous system or hormones, and then alters the animal's behavior to obtain a supply of food. These behaviors include looking for food and eating it; and regulating both food seeking and food consumption behaviors is crucial for the animal's chances of survival and reproduction.

Studies that used fruit flies as a model have previously shown that flies walk more when they are hungry. This activity helped the flies to locate and occupy food sources, but it was not clear how this food seeking behavior was regulated.

Now, Yu, Huang et al. find that a small group of neurons in the fly brain controls food seeking in starving flies. The neurons achieve this by sensing two groups of hormones with opposing activity. These hormones are the fly's equivalents of glucagon and insulin, which are found in humans and other mammals. In humans, glucagon is released when blood sugar levels are low and stimulates hunger, while insulin is released when blood sugar is high and acts to suppress feelings of hunger. Therefore, food seeking in the flies is under the precise control of signals of hunger and satiety.

Further experiments show that these fly neurons use a chemical messenger called octopamine to convey the hormone-based signals to other circuits of neurons. Notably, these downstream neurons are not involved in regulating the consumption of food. Therefore, food seeking and eating appear to be independently regulated in fruit flies.

Further studies are now needed to dissect the downstream circuits of neurons that actually control the food seeking behavior. It will also be important to explore how this behavior is suppressed when a food source is detected.

sources. In fruit flies, starvation increases the sensitivity of Or42b$^+$ olfactory receptor neurons (ORNs) that mediate odor attraction, and decreases the sensitivity of aversive Or85a$^+$ ORNs, resulting in enhanced behavioral attraction to food odors (*Ko et al., 2015*; *Root et al., 2011*). Similarly, starvation also influences the sensitivity of sweet-sensing Gr5a$^+$ gustatory receptor neurons (GRNs) and bitter-sensing Gr66a$^+$ GRNs in opposite directions, resulting in enhanced attractiveness to food taste (*Inagaki et al., 2012*; *Inagaki et al., 2014*; *Marella et al., 2012*). A number of neurohormonal cues, including DILPs, NPF, sNPF, dopamine, and tachykinin (DTK), are involved in the sensory modulation by starvation (*Inagaki et al., 2012*; *Inagaki et al., 2014*; *Ko et al., 2015*; *Marella et al., 2012*; *Root et al., 2011*).

On the other hand, starvation also promotes locomotor activity in both rodents and fruit flies, which may facilitate the exploration of the environment and increase the possibility to locate potential food sources (*Dietrich et al., 2015*; *Isabel et al., 2005*; *Lee and Park, 2004*; *Yang et al., 2015*). In fruit flies, starvation-induced hyperactivity requires the neuroendocrine cells producing AKH, the insect analog of glucagon (*Isabel et al., 2005*; *Lee and Park, 2004*). However, AKH has a role in modulating lipid storage as well as starvation resistance and whether it imposes a direct effect on starvation-induced hyperactivity remains unclear (*Bharucha et al., 2008*; *Grönke et al., 2007*). We have also shown that octopamine, the insect analog of norepinephrine, is required for starvation induced hyperactivity (*Yang et al., 2015*). But given the diverse anatomical distribution and physiological function of octopaminergic neurons in the fly brain, the underlying neural circuitry of starvation-induced hyperactivity and how it is modulated by hormonal cues remain largely unknown (*El-Kholy et al., 2015*).

Starvation-induced hyperactivity and food consumption are reciprocally inhibitory and the transition of these two behaviors relies on the detection of food cues (*Chen et al., 2015*; *Yang et al., 2015*). It is of great interest to understand how these two behaviors are dynamically regulated by the nutritional states and the availability of food sources. In rodents, acute activation of hypothalamic AgRP/NPY neurons induces both hyperactivity and food consumption, suggesting that these two behaviors are regulated by a same group of neurons and hormonal cues (*Aponte et al., 2011*; *Atasoy et al., 2012*; *Betley et al., 2015*; *Dietrich et al., 2015*; *Krashes et al., 2011*). However, the detection of food sources rapidly suppresses starvation-induced hyperactivity along with the activity

of AgRP/NPY neurons before actual food consumption, suggesting that these neurons may only be involved in the locomotor response upon starvation but not food consumption (*Betley et al., 2015*; *Chen et al., 2015*). Thus, it remains elusive whether starvation-induced hyperactivity and food consumption are modulated by the same set of hormones via a common neural target, or they are independently regulated (*Seeley and Berridge, 2015*).

In this present study, we aimed to better understand how starvation-induced hyperactivity is regulated by hormonal signals, and whether starvation-induced hyperactivity and food consumption are interdependently or independently regulated. Through a neuronal-specific RNAi screen, we find that a small group of octopaminergic neurons located in the subesophageal zone (SEZ) of the fly brain are both necessary and sufficient for starvation-induced hyperactivity. These neurons co-express AKHR and dInR, the receptor of hunger hormone AKH and the receptor of satiety hormone DILPs, respectively, which modulate starvation-induced hyperactivity in opposite directions. Thus, these AKHR$^+$dInR$^+$ octopaminergic neurons represent a common neural target for two sets of functionally antagonizing hormones to modulate starvation-induced hyperactivity. Notably, manipulating starvation-induced hyperactivity does not interfere with starvation-induced food consumption, suggesting that starvation-induced changes in different food intake behaviors are independently regulated by different hormonal cues and neural circuitry in fruit flies.

## Results

### A neuron-specific RNAi screen identified AKHR required for starvation-induced hyperactivity

We have previously shown that starvation induces sustained increase in locomotor activity of adult flies (*Yang et al., 2015*). To identify the hormonal cues that regulate this behavior, we performed a neuron-specific RNAi screen in adult flies and examined the influence on their locomotor activity, which was indirectly measured by their frequency to cross the midline of tubes in the *Drosophila* Activity Monitor System (DAMS, Trikinetics) (*Figure 1a–b*).

We crossed *UAS-RNAi* lines targeting 31 candidate neuropeptide receptors to a pan-neuronal GAL4 driver, *elav-GAL4,* and assayed their female progeny for baseline locomotion under fed conditions (*Figure 1c*). 16 out of the 31 lines exhibited significantly altered activity compared to the control, suggesting that the baseline locomotion may be a behavioral trait sensitive to multiple neuropeptidergic signaling systems (*Supplementary file 1*).

Besides the baseline activity, we also examined the changes in locomotor activity upon starvation (*Figure 1c*). Consistent with our previous report (*Yang et al., 2015*), control flies exhibited ~60% increase in their locomotor activity upon starvation (*Figure 1d*, 'Control'). Amongst 31 RNAi lines we examined, only the neuronal knock-down of AKHR eliminated starvation-induced hyperactivity (*Figure 1d*, 'AKHR'), whereas all other RNAi lines still exhibited significantly enhanced locomotion upon starvation (*Figure 1d*). Notably, neuronal knock-down of AKHR did not interfere with the baseline locomotion under fed conditions (*Supplementary file 1*), suggesting that AKHR is not involved in general motor control, but specifically in the regulation of locomotion by the internal nutritional states.

Consistent with the results from our RNAi screen, we found that *AKHR*$^{-/-}$ mutants exhibited no increase in locomotion upon starvation (*Figure 2a–c*). It is worth noting that a previous report did not observe the behavioral difference between *AKHR*$^{-/-}$ mutants vs. the control under starvation conditions, which was likely due to the short time window of the behavioral assay (*Bharucha et al., 2008*).

AKHR is the candidate receptor of AKH, the insect analog of mammalian glucagon. Similar to its mammalian counterpart, AKH secretion is induced by the reduction in circulating sugar levels, which in turn mobilizes lipid storage for energy supply (*Kim and Rulifson, 2004*). Previous reports have shown that genetic ablation of AKH-producing cells located in the corpora cardiaca led to increased fat storage and diminished starvation-induced hyperactivity (*Isabel et al., 2005*; *Lee and Park, 2004*). Consistently, we also found that eliminating AKH expression abolished starvation-induced hyperactivity (*Figure 2d–f*). Taken together, AKH-AKHR signaling is required for starvation-induced hyperactivity in adult flies.

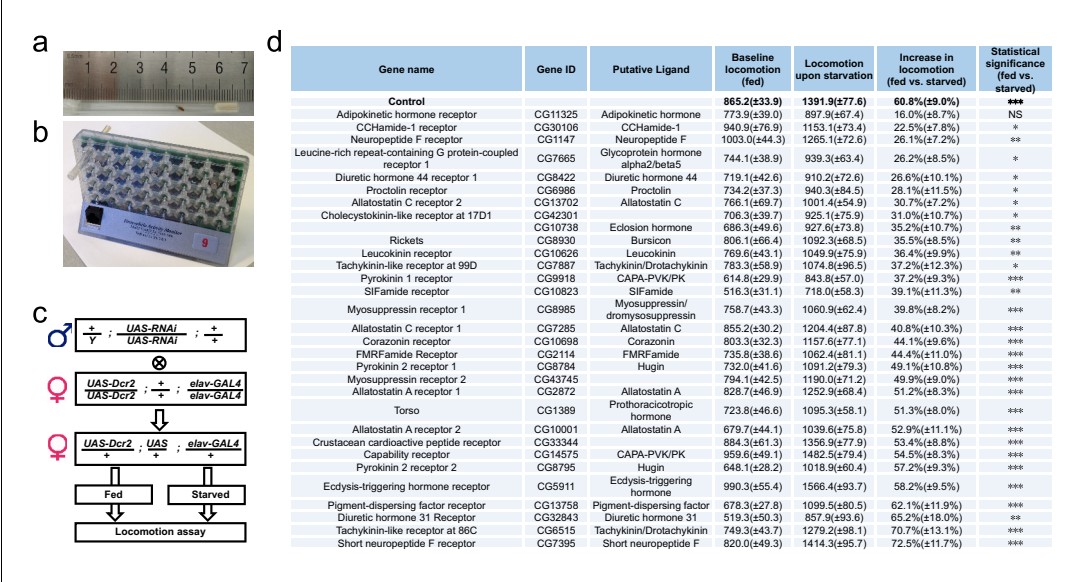

**Figure 1.** A neuron-specific RNAi screen for starvation-induced hyperactivity in adult *Drosophila*. (a–b) The DAMS-based locomotion assay. (a) A single virgin female fly was hosted in a 5 × 65 mm polycarbonate tube, secured between 2% agar medium (with or without 5% sucrose) on one end and a piece of cotton on the other. The ruler was used for illustrating the size of the tube. (b) The tube was then inserted into the DAMS monitor (Trikinetics). One DAM2 monitor can hold 32 tubes at a time. The passage of flies through the middle of the tube was counted by an infrared beam. The frequency of midline crossings therefore indirectly measured flies' locomotor activity. (c) The workflow of our RNAi screen. Note that *UAS-RNAi* transgenes were integrated into either the second or the third chromosome. (d) Summary of our RNAi screen (n = 29–48). For each line, their average daily midline crossing activity when fed *ad libitum* with 5% sucrose ('Baseline locomotion'), their activity upon starvation ('Locomotion upon starvation'), and the relative increase in locomotion ('Increase in locomotion') were listed. 'Control' is the progeny of GAL4 driver line crossed to a wild type strain. The statistical difference between fed vs. starved conditions were listed for each RNAi line. Data are shown as means ( ± SEM). NS, p>0.05; *p<0.05; **p<0.01; ***p<0.001.

## Neuronal AKHR is required for starvation-induced hyperactivity but not food consumption

Previous reports have shown that AKHR is expressed in the fat body and regulates lipid storage (*Bharucha et al., 2008*; *Grönke et al., 2007*). We also confirmed that $AKHR^{-/-}$ mutant flies had excessive fat storage (*Figure 3—figure supplement 1*). Therefore, it was unclear whether AKHR signaling had a direct role in regulating starvation-induced hyperactivity in the nervous system, or merely delayed starvation-induced hyperactivity by enhancing lipid storage, or both. To further examine the function of neuronal AKHR, we used a different pan-neuronal driver *neuronal synaptobrevin-GAL4 (nSyb-GAL4)* to knock down AKHR in the nervous system. Consistent with the results from our RNAi screen, pan-neuronal elimination of AKHR showed comparable locomotor activity under fed and starved conditions, whereas the two genetic controls exhibited significant increase in locomotion upon starvation (*Figure 3a–c*). Notably, neuronal knock-down of AKHR did not affect lipid storage of flies (*Figure 3—figure supplement 1*), and fat body knock-down of AKHR did not eliminate starvation-induced hyperactivity (*Figure 3—figure supplement 2*), disassociating the behavioral and metabolic effects of AKHR signaling. These results confirm that only neuronal AKHR, but not fat body-expressed AKHR, is required for starvation-induced hyperactivity in flies.

To further evaluate the function of neuronal AKHR in starvation-induced hyperactivity, we used a behavioral assay that directly quantified the walking velocity as well as the position of individual flies in the presence or absence of food cues (*Figure 3—figure supplement 3a*). By using this behavioral assay, we confirmed that flies with neuronal knock-down of AKHR exhibited comparable walking velocity under fed vs. starved conditions, whereas the two control genotypes exhibited significantly increased walking velocity upon starvation (*Figure 3—figure supplement 3e*). Besides increased locomotion, starved flies also exhibited increased interest to food and could rapidly locate and occupy food sources (*Ko et al., 2015*; *Root et al., 2011*; *Yang et al., 2015*). Interestingly, starved

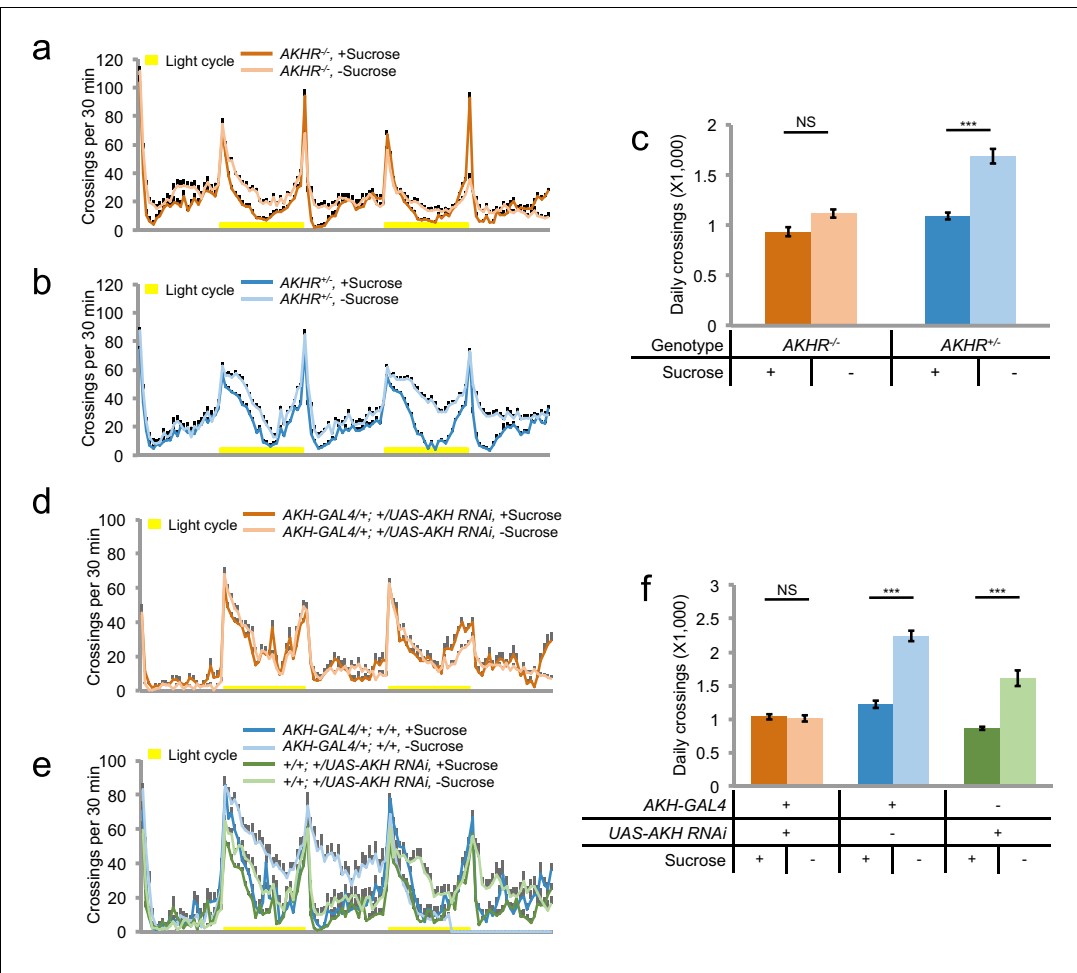

**Figure 2.** AKH-AKHR signaling is required for starvation-induced hyperactivity. (**a–b**) Midline crossing activity of indicated genotypes assayed in the presence of 5% sucrose ('+Sucrose') or 2% agar ('+Sucrose') (n = 61–63). Yellow bars represent 12 hr light-on period in this and following figures. (**c**) Average daily midline crossing activity of flies assayed in **a–b**. (**d–e**) Midline crossing activity of indicated genotypes assayed in the presence of 5% sucrose ('+Sucrose') or 2% agar ('-Sucrose') (n = 24–30). (**f**) Average daily midline crossing activity of flies assayed in **d–e**. Error bars represent SEM. NS, p>0.05; ***p<0.001.

flies from all three genotypes could locate and occupy food sources in the center of the behavioral chamber (**Figure 3—figure supplement 3b–d**). Taken together, these data confirm that neuronal AKHR is required for starvation-induced hyperactivity without interfering their ability to locate and occupy food sources.

Starvation promotes both locomotor activity and food consumption. But it remained unclear whether these two behaviors are regulated by the same set of hormonal cues or not. We thus asked whether neuronal AKHR was also required for starvation-induced increase in food consumption. We first examined food consumption during the course of a single meal, by using a capillary-based feeding assay named the MAFE (MAnual FEeding) assay (**Qi et al., 2015**) (**Figure 3d**). We found that starvation could significantly increase food consumption, and that neuronal knock-down of AKHR did not affect starvation-induced increase in meal size (**Figure 3f**).

We also assayed long-term food consumption using the FLIC (Fly Liquid-food Interaction Counter) assay that measured the frequency and duration of physical contacts between fly's proboscis and the liquid food (**Ro et al., 2014**) (**Figure 3e**, also see **Figure 3g–i**). We found that eliminating AKHR in neurons did not interfere with long-term feeding, as measured by both the number of feeding bouts and the total duration of feeding (**Figure 3j–k**). Taken together, these data show that neuronal

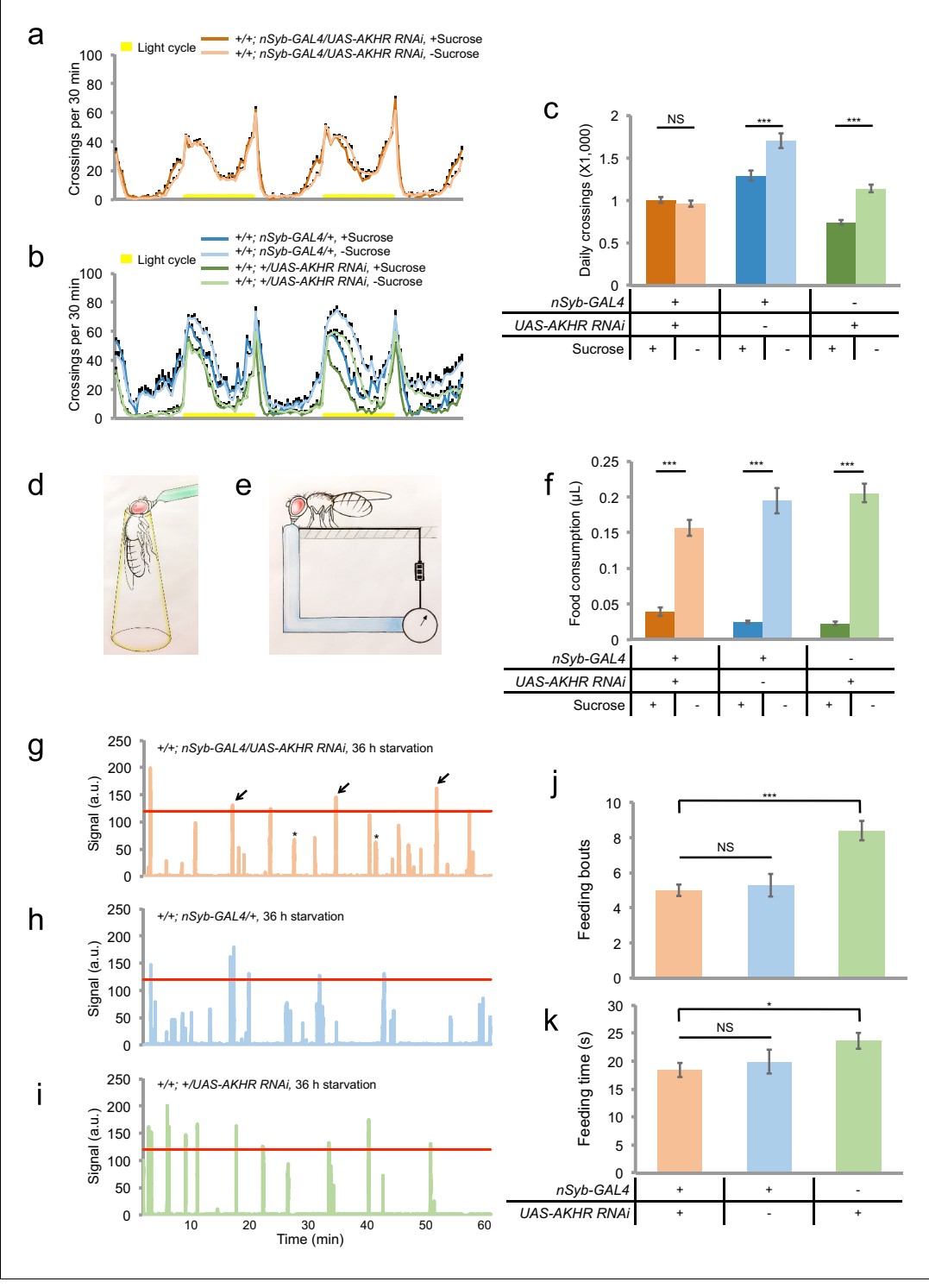

**Figure 3.** Neuronal AKHR is required for starvation-induced hyperactivity but not food consumption. (**a–b**) Midline crossing activity of indicated genotypes assayed in the presence of 5% sucrose ('+Sucrose') or 2% agar ('-Sucrose') (n = 31–80). (**c**) Average daily midline crossing activity of flies assayed in **a–b**. (**d–e**) Schematic illustration of the MAFE assay (**d**) or the FLIC assay (**e**). (**f**) Volume of 800 mM sucrose consumed in a meal by indicated genotypes fed *ad libitum* with 5% sucrose, or starved for 36 hr using the MAFE assay. (n = 30–39). (**g–i**) Representative feeding plots of individual flies in the FLIC assay. Flies of the indicated genotypes were starved for 36 hr before the assays. The plots show the electrical current signals that reflected the physical contact between flies and the liquid food (800 mM sucrose). The a.u. higher than 120 (red line) was considered as feeding events (arrows).
*Figure 3 continued on next page*

*Figure 3 continued*

Asterisks indicate possible 'tasting/licking' events. (j–k) Number of feeding bouts (j) and total duration of feeding time (k) during 1 hr recording in the FLIC assay (n = 50–72). Note that one control genotype (green) exhibited higher levels of feeding likely due to the genetic background of *nSyb-GAL4* we used. Error bars represent SEM. NS, p>0.05; *p<0.05; ***p<0.001.

The following figure supplements are available for figure 3:

**Figure supplement 1.** Neuronal AKHR is not involved in the regulation of fat storage.

**Figure supplement 2.** AKHR expressed in the fat body is not required for starvation-induced hyperactivity.

**Figure supplement 3.** Neuronal AKHR is required for starvation-induced hyperactivity but not the localization and occupation of food sources.

AKHR is specifically required for starvation-induced hyperactivity but not food consumption. It is likely that several other hormonal cues mediate the effect of starvation on food consumption (Figure 7i) (*Pool and Scott, 2014*).

## AKHR[+] neurons are required for starvation-induced hyperactivity

We next asked which AKHR[+] neurons were required for starvation-induced hyperactivity. Previous reports have shown that besides the fat body, AKHR is also expressed in the sweet-sensing gustatory neurons (*Bharucha et al., 2008*). However, we found that knocking down AKHR in Gr5a[+] sweet-sensing gustatory neurons did not interfere with starvation-induced hyperactivity (*Figure 4—figure supplement 1*). Moreover, we have previously shown that starvation-induced hyperactivity did not require gustatory input (*Yang et al., 2015*). These data suggest that AKHR is likely expressed in additional neurons other than those in the gustatory sensory organs, where it regulates starvation-induced hyperactivity.

To specifically examine the neuronal expression of AKHR, we utilized the split-GAL4 system and generated transgenic flies carrying the GAL4 Activation Domain under the control of nSyb promoter (nSyb:AD) and the GAL4 DNA-binding Domain under the control of AKHR promoter (AKHR:BD). Using the combinational nSyb:AD/AKHR:BD-GAL4 driver that induces gene expression only in AKHR[+] neurons, we found that AKHR was expressed in a small group of neurons located in the SEZ (2–4 neurons per hemisphere) but not in the ventral nerve cord (data not shown) (*Figure 4a–b*). The cell bodies of these AKHR[+] neurons were located in the ventrolateral region of the SEZ and sent Y-shaped neural projections into the SEZ (*Figure 4b*). As previously reported (*Bharucha et al., 2008*), we also found that AKHR was expressed in the sweet-sensing gustatory neurons and projected to the SEZ region of the brain (*Figure 4c–d*). As discussed above, however, these gustatory neurons are not involved in starvation-induced hyperactivity (*Figure 4—figure supplement 1*).

To test whether AKHR[+] neurons in the SEZ were required for starvation-induced hyperactivity, we expressed Shibire[ts1], a temperature sensitive form of dynamin (*Kitamoto, 2001*), in AKHR[+] neurons and acutely blocked neuronal transmission by transferring the flies to non-permissive temperature (30°C) before the behavioral assays. We found that acute silencing of AKHR[+] neurons eliminated starvation-induced hyperactivity (*Figure 4e–g*). Therefore, AKHR[+] neurons in the fly brain are required for increased locomotion upon starvation.

We also examined whether AKHR[+] neurons were required for starvation-induced increase in food consumption. Acute silencing of AKHR[+] neurons did not block the increase in food consumption by starvation, as assayed by the changes in meal size upon starvation (*Figure 4h*), and by the number of feeding bouts and total feeding duration (*Figure 4i–j*). Therefore, AKHR and AKHR[+] neurons specifically regulate starvation-induced hyperactivity, but not food consumption.

## Activation of AKHR[+] neurons promotes starvation-induced hyperactivity

We then sought to investigate whether AKHR[+] neurons played an instructive or merely permissive role in regulating starvation-induced hyperactivity. To do so, we expressed NaChBac

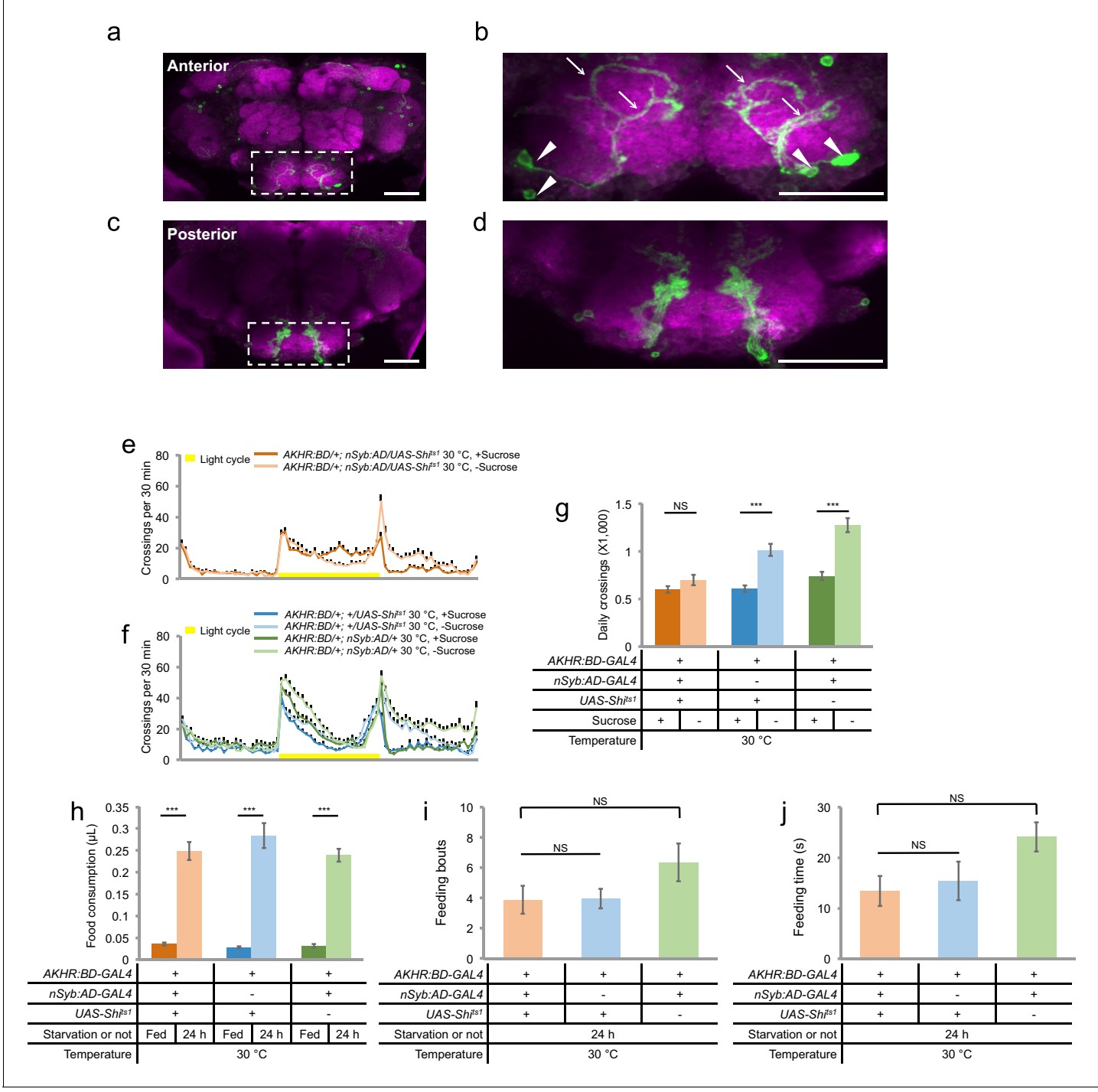

**Figure 4.** AKHR[+] neurons are required for starvation-induced hyperactivity. (**a, c**) The expression of membrane-bound GFP (mCD8GFP) in AKHR[+]neurons in the anterior (**a**) and the posterior (**c**) part of the fly brain. The dashed box outlines the region of SEZ. (**b, d**) An enlarged image of the SEZ region seen in **a** and **c**. Note the cell bodies located in the ventrolateral side of the SEZ (arrowheads) and their Y-shaped projections (arrows) in **b**. Green: GFP. Magenta: nc82. Scale bars represent 20 μm in **a–d**. (**e–f**) Midline crossing activity of indicated genotypes assayed in the presence of 5% sucrose ('+Sucrose') or 2% agar ('-Sucrose') (n = 60–75). (**g**) Average daily midline crossing activity of flies assayed in **e–f**. (**h**) Volume of 800 mM sucrose consumed in a meal by indicated genotypes fed *ad libitum* with 5% sucrose, or starved for 24 hr using the MAFE assay (n = 18–40). (**i–j**) Number of feeding bouts (**i**) and total duration of feeding time (**j**) during 1 hr recording in the FLIC assay (n = 20–24). Error bars represent SEM. NS, p>0.05; ***p<0.001.

The following figure supplement is available for figure 4:

*Figure 4 continued on next page*

*Figure 4 continued*

**Figure supplement 1.** AKHR expressed in Gr5a$^+$ gustatory sensory neurons is not required for starvation-induced hyperactivity.

(*Nitabach et al., 2006*), a bacterial sodium channel, in AKHR$^+$ neurons. NaChBac expression increases the membrane excitability and facilitates the activation of these neurons (*Nitabach et al., 2006*). We found that NaChBac expression in AKHR$^+$ neurons accelerated the onset of hyperactivity upon starvation. While the two control lines only exhibited a significant increase in locomotion during Day two of the assay, NaChBac expression in AKHR$^+$ neurons led to increased locomotor activity from Day one of the assay (*Figure 5a–d*). Consistently, over-expression of AKHR in AKHR$^+$ neurons also led to earlier onset of starvation-induced hyperactivity (*Figure 5—figure supplement 1*).

We also examined whether activation of AKHR$^+$ neurons influenced starvation-induced increase in food consumption. As shown in *Figure 5e*, NaChBac expression in AKHR$^+$ neurons did not change the meal size of both fed and starved flies. Activating AKHR$^+$ neurons did not alter the duration and frequency of feeding behavior, either (*Figure 5f–g*). Taken together, the regulation of starvation-induced hyperactivity can therefore be disassociated from the regulation of starvation-induced food consumption, since the former is neither necessary nor sufficient for the latter (*Figures 4* and *5*).

## Insulin signaling suppresses starvation-induced hyperactivity via AKHR$^+$ neurons

We next sought to further investigate the mechanism underlying the regulation of starvation-induced hyperactivity by AKHR$^+$ neurons. We hypothesized that besides AKH, AKHR$^+$ neurons might also sense other hormonal cues that exerted modulatory effects on the activity of AKHR$^+$ neurons and hence starvation-induced hyperactivity. To examine this hypothesis, we labeled AKHR$^+$ neurons with mCD8GFP, extracted individual GFP$^+$ neurons (and randomly picked GFP$^-$ neurons as negative controls) from fly brains, and performed single-cell RT-PCR analysis. All GFP$^+$ neurons we examined were AKHR positive, confirming that the nSyb:AD/AKHR:BD-GAL4 driver reliably recapitulates the endogenous expression pattern of AKHR (*Figure 6a–b*, 'GFP' and 'AKHR'). We also performed single-cell RNA-seq using the cells labeled by the nSyb:AD/AKHR:BD-GAL4 driver. Consistent with our single-cell RT-PCR experiments, all GFP$^+$ cells we examined by RNA-seq showed considerable AKHR expression (*Figure 6c*, 'AKHR'). In contrast, these GFP$^+$ cells showed no or very low expression for genes specifically expressed in the peripheral nervous system, muscle, and the fat body (*Figure 6c*, 'Or67d', 'Gr5a', 'MyoD', and 'Slimfast').

We then examined the expression of a collection of candidate neuropeptide receptors in AKHR$^+$ neurons. Among all receptor genes we examined (*Supplementary file 1*), we found that the transcripts of dInR were steadily detected in GFP$^+$ neurons by single-cell RT-PCR (*Figure 6a–b*, 'dInR'). The co-localization of dInR and AKHR was also confirmed by RNA-seq (*Figure 6c*, 'dInR') and antibody staining (*Figure 6—figure supplement 1*).

dInR is the receptor of DILPs, the insect analogs of mammalian insulin. Similar to insulin, DILPs function as satiety signals. Dietary sugars and proteins induce the release of DILPs into the hemolymph, which in turn promotes growth and nutrient storage (*Buch et al., 2008*; *Rulifson et al., 2002*). Therefore, the expression of dInR in AKHR$^+$ neurons suggests that insulin signaling may counteract the effect of AKH-AKHR signaling and suppress the activity of AKHR$^+$ neurons hence starvation-induced hyperactivity. To test this hypothesis, we knocked down the expression of dInR in AKHR$^+$ neurons and examined its effect on flies' locomotion. We found that similar to the effect of activating AKHR$^+$ neurons (*Figure 5a–d*), eliminating dInR expression in AKHR$^+$ neurons accelerated the onset of hyperactivity upon starvation (*Figure 6d–g*). Conversely, over-expression of dInR in AKHR$^+$ neurons delayed the onset of starvation-induced hyperactivity (*Figure 6—figure supplement 2*). Collectively, these data suggest that insulin signaling in AKHR$^+$ neurons suppress the activity of these neurons and hence starvation-induced hyperactivity. To confirm this finding, we over-expressed *Drosophila* PTEN (dPTEN), a negative regulator of insulin signaling pathway, in AKHR$^+$ neurons (*Gao et al., 2000*). Over-expression of dPTEN led to an earlier onset of starvation-induced hyperactivity, phenocopying the effect of dInR knock-down in AKHR$^+$ neurons (*Figure 6h–k*).

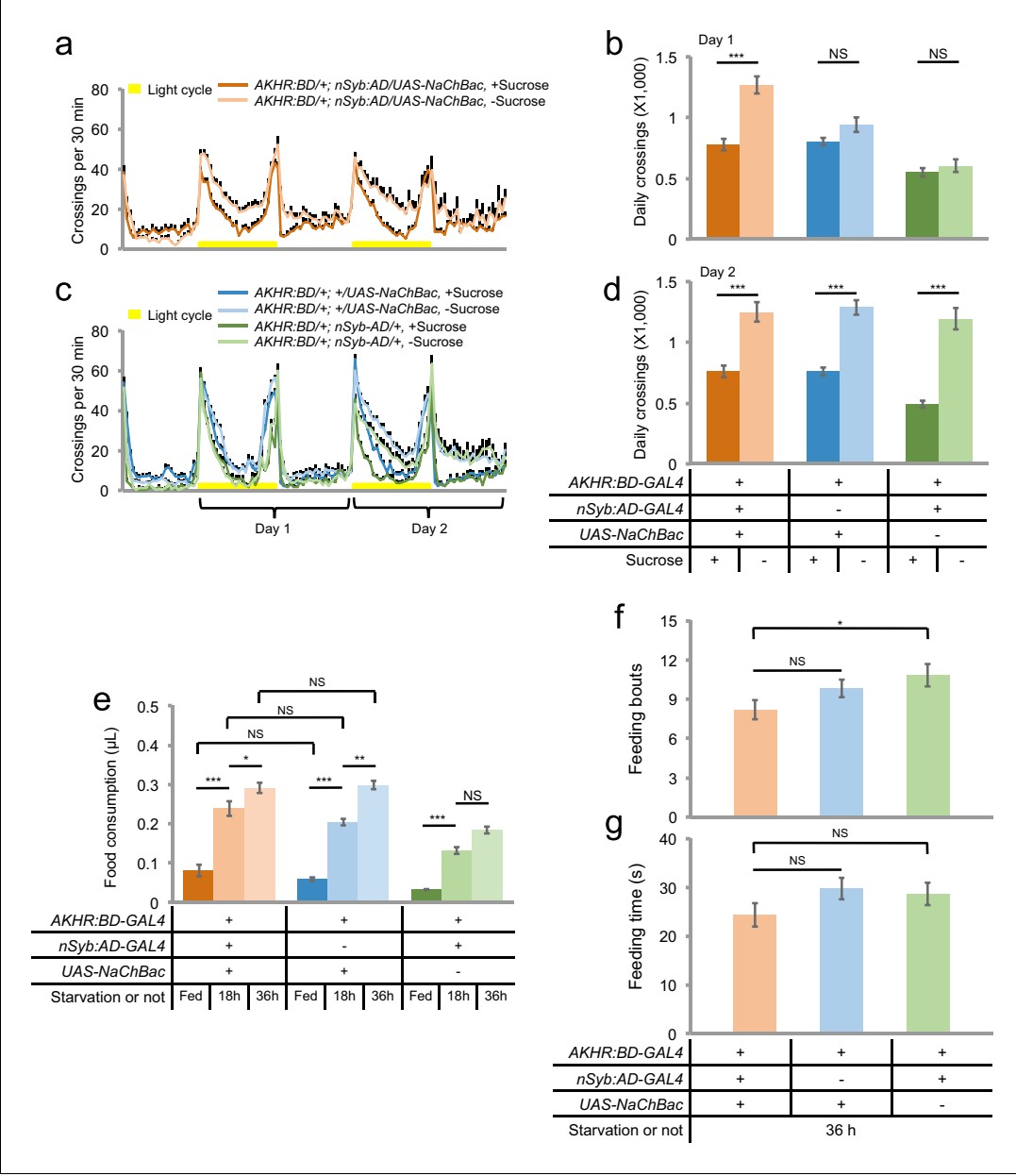

**Figure 5.** AKHR⁺ neurons promote starvation-induced hyperactivity. (**a, c**) Midline crossing activity of indicated genotypes assayed in the presence of 5% sucrose ('+Sucrose') or 2% agar ('-Sucrose') (n = 48–80). (**b, d**) Average daily midline crossing activity during Day one (**b**) and Day two (**d**) of flies assayed in **a, c**. (**e**) Volume of 800 mM sucrose consumed in a meal by indicated genotypes fed *ad libitum* with 5% sucrose, or starved for 18 and 36 hr using the MAFE assay (n = 38–52). (**f–g**) Number of feeding bouts (**f**) and total duration of feeding time (**g**) during 1 hr recording in the FLIC assay (n = 54–60). Error bars represent SEM. NS, p>0.05; *p<0.05; **p<0.01; ***p<0.001.

The following figure supplement is available for figure 5:

**Figure supplement 1.** AKHR over-expression in AKHR⁺ neuron accelerates starvation-induced hyperactivity.

Taken together, these data suggest that the activity of AKHR⁺ neurons is modulated by two sets of functionally antagonizing hormones, AKH and DILPs. The hunger hormone AKH activates these neurons via AKHR and promotes locomotor activity upon starvation, whereas the satiety hormones DILPs exert a suppressive effect on starvation-induced hyperactivity via dInR signaling. The interplay

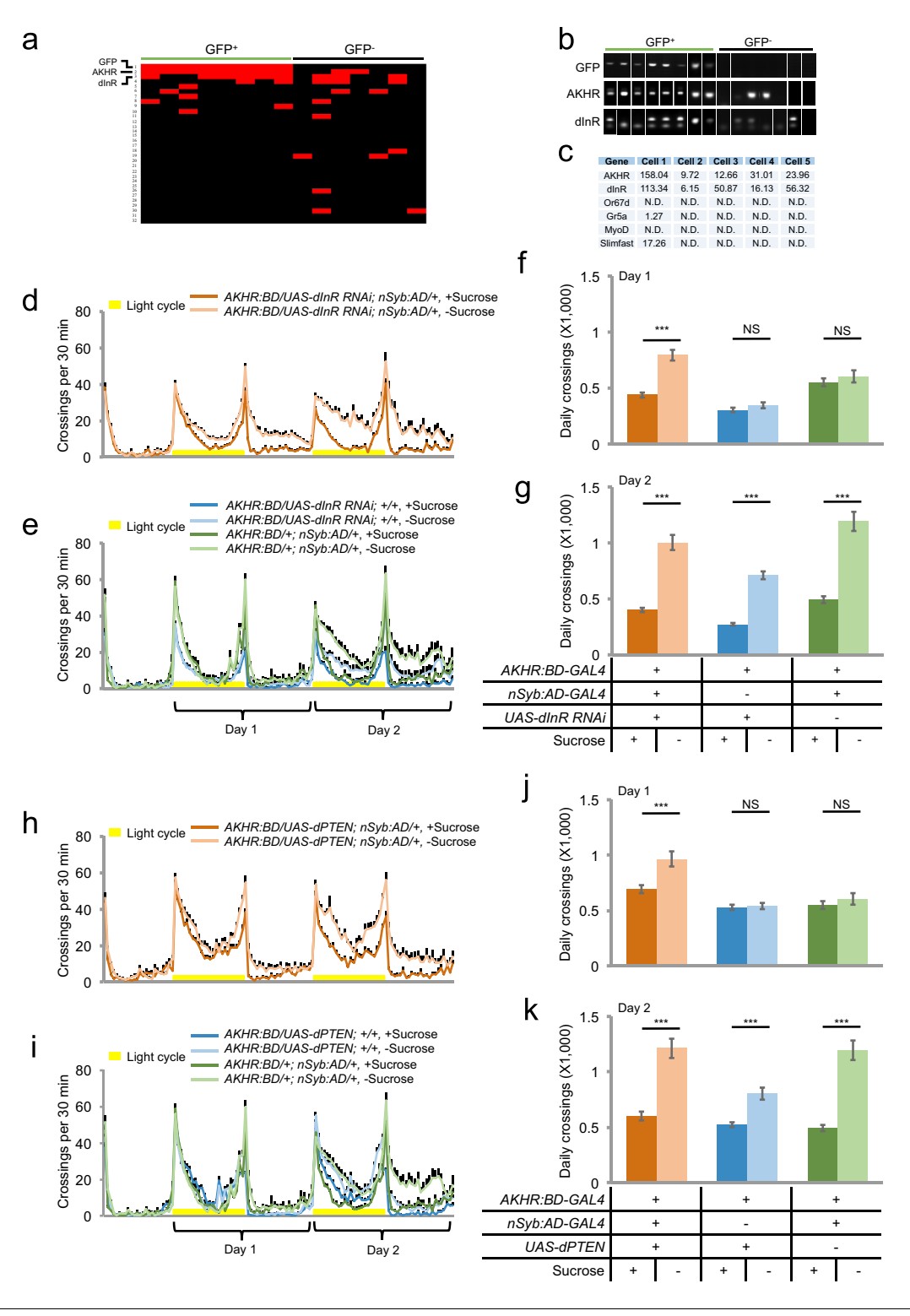

**Figure 6.** Insulin signaling suppresses starvation-induced hyperactivity via AKHR[+] neurons. (**a**) Heat map indicating the expression pattern of neuropeptide receptors we examined in individual AKHR[+] (green) and AKHR[−] (black) neurons. Red and black blocks represent genes that could and could not be detected by RT-PCR, respectively. For a complete list of all 32 genes we examined see *Supplementary file 1*. The first three genes, GFP, AKHR, and dlnR are indicated in bold. (**b**) Representative RT-PCR bands for indicated genes shown in **a**. Note that the lower bands in the 'dlnR' row are primer dimers. (**c**) Gene expression in five GFP[+] cells assayed by RNA-seq (shown in *Figure 6 continued on next page*

*Figure 6 continued*

RPKM). Or67d and Gr5a are expressed in the primary sensory neurons; MyoD in muscle; and Slimfast in the fat body. (**d–e**) Midline crossing activity of indicated genotypes assayed in the presence of 5% sucrose ('+Sucrose') or 2% agar ('-Sucrose') (n = 48–80). (**f–g**) Average daily midline crossing activity during Day one and Day two of flies assayed in **d–e**. (**h–i**) Midline crossing activity of indicated genotypes assayed in the presence of 5% sucrose ('+Sucrose') or 2% agar ('-Sucrose') (n = 48–80). (**j–k**) Average daily midline crossing activity during Day one and Day two of flies assayed in **h–i**. Error bars represent SEM. NS, p>0.05; ***p<0.001.

The following figure supplements are available for figure 6:

**Figure supplement 1.** Co-localization of AKHR and dInR in the fly brain.
**Figure supplement 2.** dInR over-expression in AKHR⁺ neuron delays starvation-induced hyperactivity.
**Figure supplement 3.** DMS-R2 in AKHR⁺ neurons is not required for starvation-induced hyperactivity.

between these two sets of hormonal cues therefore modulates locomotor activity upon changes in flies' internal nutritional states. Besides dInR, Myosuppressin receptor 2 (DMS-R2) was also moderately enriched in AKHR⁺ neurons (*Figure 6a*, Line 4). Knocking down DMS-R2 in AKHR⁺ neurons, however, did not affect starvation-induced hyperactivity (*Figure 6—figure supplement 3*).

## Octopamine signaling mediates the effect of AKHR⁺ neurons on starvation-induced hyperactivity

We next asked how AKHR⁺ neurons connected the upstream hormonal signals to downstream neural circuitry. To do so, we aimed to understand which neurotransmitter(s) AKHR⁺ neurons utilized to modulate starvation-induced hyperactivity. We performed single-cell RT-PCR on AKHR⁺ neurons for genes critical for the biosynthesis of several neurotransmitters including dopamine, serotonin, octopamine, tyramine, acetylcholine, glutamate, GABA, and histamine (*Supplementary file 1*). Single-cell RT-PCR showed that the transcripts of tyramine beta-hydroxylase (TβH) were detected in GFP⁺ neurons (*Figure 7a–b*). These results were also confirmed by single-cell RNA-seq (*Figure 7c*).

TβH is critical for the biosynthesis of octopamine, the insect analog of vertebrate norepinephrine (*Roeder, 2004*). Octopamine is involved in the regulation of various behaviors, including sleep (*Crocker and Sehgal, 2008*), learning (*Burke et al., 2012*), and aggression (*Hoyer et al., 2008*). We have also shown previously that octopamine signaling was also required for starvation-induced hyperactivity (*Yang et al., 2015*). We therefore asked whether octopamine mediated the effect of AKHR⁺ neurons on this behavior. We first showed that similar to the phenotype of TβH^M18 null mutants (*Yang et al., 2015*), pan-neuronal knock-down of TβH expression eliminated starvation-induced hyperactivity (*Figure 7—figure supplement 1*). Furthermore, knocking down the expression of TβH specificallyin AKHR⁺ neurons eliminated starvation-induced hyperactivity (*Figure 7d–f*). Consistently, knocking down the expression of AKHR in octopaminergic neurons also blocked the increase in locomotion by starvation (*Figure 7g*). Taken together, our data suggest that AKHR⁺ neurons in the fly brain are octopaminergic, and that octopamine signaling mediates the effect of these neurons on starvation-induced hyperactivity.

Single-cell RT-PCR also identified that the transcripts of Dopa decarboxylase (DDC), a key enzyme for the biosynthesis of both dopamine and serotonin (*Lundell and Hirsh, 1994*), was moderately enriched in AKHR⁺ neurons (*Figure 7a–b*). We thus examined the potential role of DDC in AKHR⁺ neurons. Nevertheless, eliminating DDC expression in AKHR⁺ neurons exerted no effect on the increased locomotion by starvation (*Figure 7h*).

## Discussion

Food seeking and food consumption are essential for the acquisition of food sources, and hence survival, growth, and reproduction of animal species (*Gao and Horvath, 2007*). Starvation influences food-seeking behavior via both modulating the perception of food cues as well as enhancing flies' locomotor activity (*Figure 7i*). Accumulated evidence has suggested that starvation modulates the

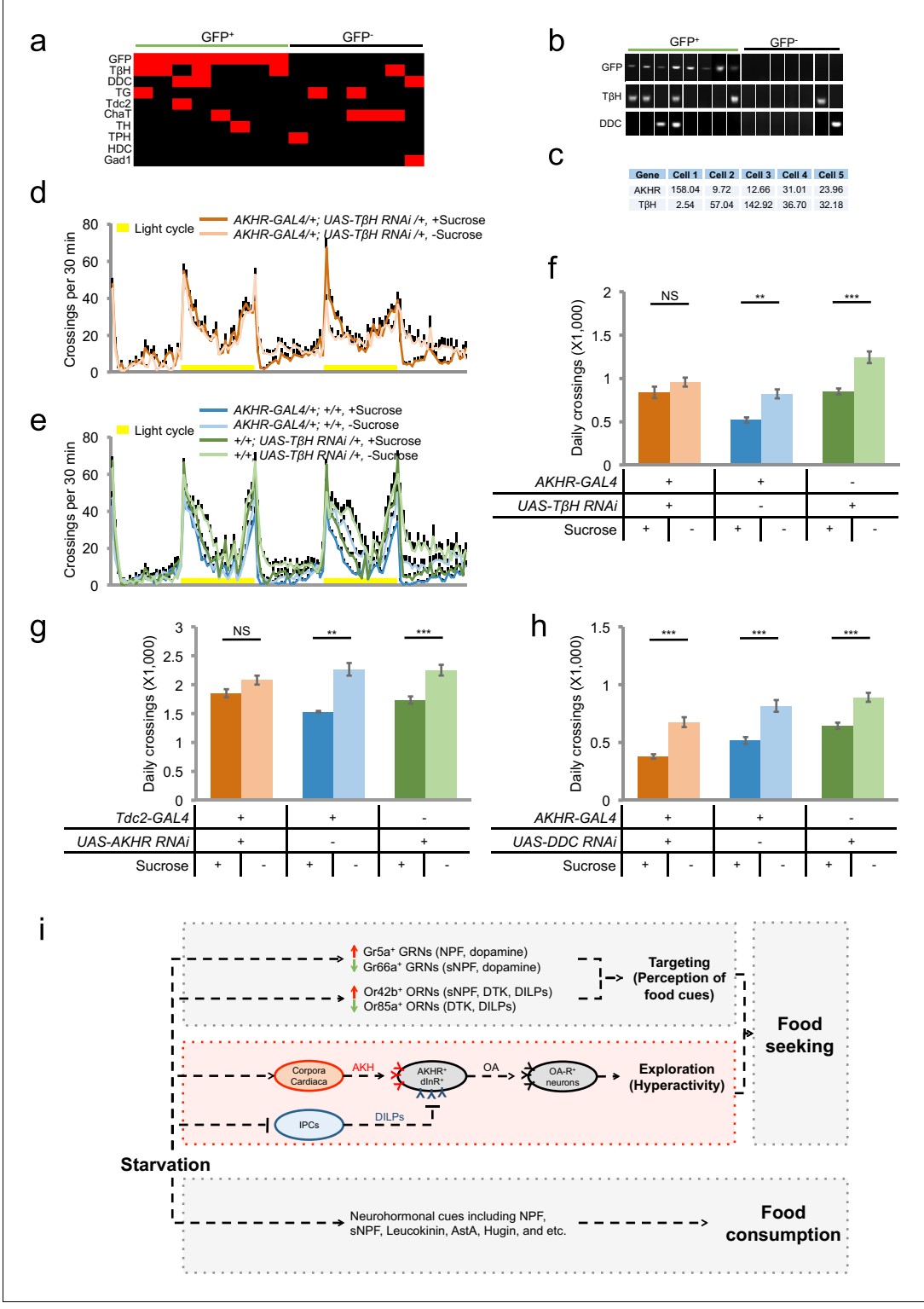

**Figure 7.** Octopamine mediates the effect of AKHR⁺ neurons on starvation-induced hyperactivity. (**a**) Heat map indicating the expression of neurotransmitter related genes in individual AKHR⁺ and AKHR⁻ neurons. Red and black blocks represent genes that could and could not be detected by RT-PCR, respectively. For a complete list of all 10 genes we examined see **Supplementary file 1**. (**b**) Representative RT-PCR bands for indicated genes shown in **a**. (**c**) Gene expression in five GFP⁺ cells assayed by RNA-seq (shown in RPKM). (**d–e**) Midline crossing activity of indicated genotypes assayed in the presence of 5% sucrose ('+Sucrose') or 2% agar ('-Sucrose') (n = 38–48). (**f**) Average daily midline crossing activity of indicated genotypes in **d–e**. (**g–h**) Average daily midline crossing activity

*Figure 7 continued on next page*

*Figure 7 continued*

of indicated genotypes (n = 43–65). (i) A working model. Starvation promotes both food seeking and food consumption. Food seeking has two components: food targeting (i.e. perception of food cues) and environmental exploration (i.e. hyperactivity). In this present study (highlighted in red), we have shown that a small group of octopaminergic neurons located in the fly brain are both necessary and sufficient for starvation-induced hyperactivity, an important aspect of food seeking. These neurons express AKHR and dInR, the receptors of AKH and DILPs, respectively. These neurons are octopaminergic (OA), and likely exert their behavioral effect via downstream neurons expressing certain octopamine receptor(s) (OA-R). It is worth noting that the regulation of starvation-induced hyperactivity is independent from that of food consumption, and vice versa. NS, p>0.05; **p<0.01; ***p<0.001.

The following figure supplement is available for figure 7:

**Figure supplement 1.** TβH is required for starvation-induced hyperactivity.

activity of ORNs via multiple neural and hormonal cues (*Beshel and Zhong, 2013*; *Ko et al., 2015*), which in turn facilitates odor-driven food search (*Root et al., 2011*) and food consumption (*Farhadian et al., 2012*; *Wang et al., 2013*). Similarly, starvation also modulates the perception of food taste via the relative sensitivity of appetitive sweet-sensing and aversive bitter-sensing GRNs, which may in turn increase the attractiveness of food taste (*Inagaki et al., 2012*; *Inagaki et al., 2014*; *Marella et al., 2012*). However, how starvation increases the locomotor activity of flies remains largely uncharacterized.

Consistent with previous reports (*Isabel et al., 2005*; *Lee and Park, 2004*), we have shown that starved fruit flies exhibit sustained increase in their locomotor activity, which can be suppressed by food consumption induced by both nutritive and non-nutritive food cues (*Yang et al., 2015*). In the present study, we have shown that a small group of neurons located in the SEZ region of the fly brain are both necessary and sufficient for starvation-induced hyperactivity. These neurons sense the changes in flies' internal nutritional states by directly responding to two sets of hormones, AKH and DILPs, and modulate locomotor activity in response. Single cell analysis has identified that these $AKHR^+dInR^+$ neurons are octopaminergic, which offers an entry point to trace the downstream neural circuitry that regulates starvation-induced hyperactivity. For example, there are 7 candidate octopamine receptors in fruit flies and it would be of interest to investigate whether any of these receptors and the receptor-expressing neurons are involved in locomotor regulation upon starvation (*El-Kholy et al., 2015*).

AKH and DILPs are two sets of functionally counteracting hormones in fruit flies. As its mammalian analog glucagon, the reduction in circulating sugars induces the release of AKH, which in turn mobilizes fat storage and provides energy supply for flies (*Bharucha et al., 2008*; *Kim and Rulifson, 2004*; *Lee and Park, 2004*). In contrast, DILPs, the insect analog of mammalian insulin, function as satiety hormones. Dietary nutrient induces the release of DILPs into the hemolymph, which in turn promotes protein synthesis, body growth, and other anabolic processes (*Buch et al., 2008*; *Rulifson et al., 2002*). We have shown that these two hormonal signaling systems exert opposite effects on starvation-induced hyperactivity via a small group of $AKHR^+InR^+$ octopaminergic neurons. These results suggest that these $AKHR^+dInR^+$ neurons can integrate the inputs from the two hormonal signaling systems representing hunger and satiety at the same time, and modulate flies' locomotor activity (*Figure 7i*). This elegant yet concise design allows these neurons to be responsive to rapid changes in the internal nutritional states as well as food availability. Furthermore, it is possible that besides hunger and satiety, other physiological states such as wakefulness, stress, and emotions also influence flies' locomotor activity. Notably, our single cell analysis has shown that these $AKHR^+$-$dInR^+$ neurons also sparsely express other neuropeptide receptors, suggesting that at least small portions of these neurons may also receive input from other neuropeptidergic systems.

Starved animals exhibited increased locomotion and food consumption, the transition of which relies on the detection of food cues (*Chen et al., 2015*; *Yang et al., 2015*). But whether these two behaviors are interdependently or independently regulated remains unclear. In this present study, we have shown that these two behaviors are dissociable from each other in fruit flies (*Figure 7i*). On the one hand, although $AKHR^+$ neurons exert a robust modulatory effect on starvation-induced

hyperactivity, these neurons are neither necessary nor sufficient for starvation-induced food consumption (Figures 4 and 5). On the other hand, the regulation of food consumption is independent of starvation-induced hyperactivity as well. We have previously shown that a small subset of GABAergic neurons in the fly brain regulates food consumption but exerts no effect on starvation-induced hyperactivity (Pool et al., 2014). In addition, several neuropeptides are known to regulate food consumption, such as Hugin, NPF, sNPF, Leucokinin, and AstA (Al-Anzi et al., 2010; Hergarden et al., 2012; Lee et al., 2004; Melcher and Pankratz, 2005; Wu et al., 2005). However, we found in our RNAi screen that the receptors of these neuropeptides were not involved in the regulation of starvation-induced hyperactivity (Figure 1d). Taken together, it is likely that starvation-induced hyperactivity and food consumption are independently regulated by different sets of hormonal cues, and that AKHR$^+$ neurons are only involved in the former but not the latter. Our results may shed light on the regulation of food intake in mammals, especially whether starvation-induced hyperactivity and food consumption are also independently regulated by different sets of hormones and distinct neural circuitry in mammals.

## Materials and methods

### Flies

Flies were kept on a standard fly medium made of yeast, corn, and agar at 25°C and 60% humidity on a 12 hr light: 12 hr dark cycle. Virgin female flies were collected shortly after eclosion and kept in groups (20 flies per vial) for 5–6 d before experiments. For experiments involving Shibire$^{ts1}$, flies were raised at 18°C for 8–9 d before transferring to 30°C right before the behavioral assays.

All UAS-RNAi lines used in the screen (#25832, #25858, #25925, #25935, #25936, #25939, #25940, #26017, #27275, #27280, #27494, #27669, #27506, #27507, #27509, #27513, #27529, #27539, #28580, #28780, #28781, #28783, #29414, #29577, #29624, #31490, #34947, #31884, #31958, #33627, #38347), UAS-TβH RNAi (#27667), UAS-AKH RNAi (#27031), AKH-GAL4 (#25683), UAS-mCD8GFP (#32186), Tdc2-GAL4 (#9313), fat body specific ppl-GAL4 (#58768), and gustatory neuron specific Gr5a-GAL4 (#57592) were obtained from the Bloomington Drosophila Stock Center at Indiana University. UAS-DDC RNAi (#2197.N) and UAS-dlnR RNAi (#5713) were from the Tsinghua Fly Center. AKHR$^{-/-}$ and UAS-AKHR flies were from R. Kühnlein (Grönke et al., 2007).

Transgenic flies were generated using the methods described previously (Pfeiffer et al., 2010). Briefly, the nSyb promoter (−1826 to +78) and the AKHR promoter (−2804 to +55) were generated by PCR from genomic DNA and cloned into pDONR221 vector (Life Technologies), and subsequently cloned into pBPGUw (Addgene #17575), pBPp65ADZpUw (Addgene #26234), and pBPZpGAL4DBDUw (Addgene #26233) to generate nSyb-GAL4, AKHR-GAL4, nSyb:AD and AKHR:BD constructs. These cloned DNA fragments were then integrated into AttP40 (25C6), AttP2 (68A4), and VK00031 (62E1) landing sites, respectively.

### Locomotion assay

#### DAMS-based locomotion assay

As described in our earlier report (Yang et al., 2015), the locomotor activity of individual flies was indirectly monitored using DAMS (Trikinetics). Briefly, individual virgin female flies were lightly anesthetized and introduced into 5 × 65 mm polycarbonate tubes (Trikinetics). One end of these tubes was filled with 2% (wt/vol) agar ± 5% (wt/vol) sucrose and the other end was blocked by cotton wool. These polycarbonate tubes were then inserted into DAMS monitors and kept in fly incubators during the course of the experiments.

#### Video-based locomotion assay

To prepare for the assay, a small drop of 2% (wt/vol) agar medium (sucrose might be added as indicated) was placed in the center of each recording chamber (10 mm (Diameter) × 4 mm (Height)). Individual flies fed ad libitum or starved for 36 hr were gently aspirated into each chamber, and their activity was then video recorded for 8 hr and analyzed by the CADABRA software (Dankert et al., 2009; Yang et al., 2015).

## Food consumption assays

### The MAFE assay

As described previously (*Qi et al., 2015*), individual flies were introduced into a 200 µL pipette tip by gentle aspiration. The pipette tip was carefully cut using a blade to expose the proboscis. Flies were first water satiated and then presented with 800 mM sucrose added with 2% green dye (McCormick, Sparks, MD) delivered in a fine graduated capillary (VWR, #53432–604). When the flies became unresponsive to a series of 10 food stimuli, the assay was terminated and the total volume of ingested food was calculated.

### The FLIC assay

The FLIC assay was performed as described previously (*Ro et al., 2014*). Briefly, both feeding channels in the *Drosophila* Feeding Monitors (DFM) were filled with 800 mM sucrose added with 2% blue dye (McCormick). Individual starved flies were gently aspirated into each feeding arena and their feeding behavior was recorded for 1 hr. Physical contact between flies' proboscis and the liquid food generated an electrical current that was recorded by the DFMs. According to the original report, electrical current greater than 120 a.u. was considered as actual feeding, and the total feeding bouts and the duration of feeding time were calculated accordingly.

## Imaging

The brains of ice anesthetized flies were dissected in PBS and then fixed in 4% PFA on ice for at least 1 hr. Fixed brains were incubated in Penetration/Blocking Buffer (2% Triton X-100 and 10% Calf Serum in PBS) for 20–24 hr at 4°C and incubated in Dilution Buffer (0.25% Triton X-100 and 1% Calf Serum in PBS) with primary antibodies at 4°C for 20–24 hr. The samples were then washed in Washing Buffer (3% NaCl and 1% Triton X-100 in PBS) for $3 \times 30$ min at room temperature and subsequently incubated in secondary antibodies for 20–24 hr at 4°C. The brains were washed again in Washing Buffer before mounted in Fluoroshield with DAPI (Sigma-Aldrich) for confocal imaging (Nikon $60 \times$ A/1.20 WI). Antibodies were used at the following dilutions: mouse anti-nc82 (1:200, DSHB), rabbit anti-GFP (1:500, Life Technologies), mouse anti-GFP (1:500, abcam), rabbit anti-dInR (1:100, Cell Signaling Technology), Alexa Fluor 546 goat anti mouse (1:300, Life Technologies), Alexa Fluor 488 goat anti rabbit (1:300, Life Technologies), Alexa Fluor 546 goat anti rabbit (1:300, Life Technologies), and Alexa Fluor 488 goat anti mouse (1:300, Life Technologies).

## Single-cell RT-PCR

The method of single-cell cDNA preparation has been described previously (*Tang et al., 2010*). In brief, individual GFP$^+$ cells were picked with a glass micropipette in situ (pipettes were pulled from thick-walled borosilicate capillaries (BF120-69-10, Sutter Instruments) for initial tip opening of 1–2 µm under a dissecting microscope and transferred into lysate buffer (0.9 × PCR Reaction Buffer II, 1.35 mM MgCl$_2$, 0.45% NP40, 4.5 mM DTT, 0.18 U/µl SUPERase-In, 0.36 U/µl RNase inhibitor, 12.5 nM UP1 primer, 0.045 mM dNTP mix) immediately, followed by reverse transcription using oligo(dT) primers to generated first-strand cDNA. A poly(A) tail was then added to the 3′ end of the first-strand cDNA by terminal deoxynucleotidyl transferase. The cDNA was amplified by 29 cycles of PCR with universal oligo(dT) primers, and then tested by nested PCR with the primers listed in *Supplementary file 1*.

## Single-cell RNA-seq

Individual GFP$^+$ cells were harvested as described above, subjected to cDNA amplification (SMARTer Ultra Low RNA Kit for Sequencing, Clonetech), library preparation (NEBNext Ultra II DNA Library Prep Kit, NEB), and sequencing (Illumina Hiseq2500/4000 platform). Reads were subsequently mapped to *Drosophila* genome and only uniquely mapped reads were kept for further analysis. Gene expression levels were quantified and compared by RPKM (Reads Per Kilobase per Million mapped reads).

## Statistical analysis

Data presented in this study were verified for normal distribution by D'Agostino-Pearson omnibus test. Student's t-test, one-way ANOVA, and two-way ANOVA were applied for pair-wise

comparisons, comparisons among 3 or more data sets, and comparisons with more than one variant, respectively. The *post hoc* test with Bonferroni correction was performed for multiple comparisons following ANOVA.

## Acknowledgements

We thank members of the Wang lab for helpful discussions and technical assistance. We thank David Anderson, Zhefeng Gong, Ronald Kühnlein, Zhiwei Liu, Kristin Scott, Yi Rao, Zhuohua Zhang, the Bloomington Drosophila stock center at Indiana University, and the Tsinghua Fly Center for fly stocks. We thank Ziao Lin, Scott Pletcher, and Jenny Ro for help with the FLIC assay. We thank Jie He and Zuoren Wang for help with the single cell analysis. Danping Chen provides scientific and administrative support in the laboratory. This study was funded by the Thousand Young Talents Plan, the National Natural Science Foundation of China (No. 31522026), and the Fundamental Research Funds for the Central Universities (No. 2016QN81010) and the Zhejiang Natural Science Funds (No. LR15C060001).

## Additional information

### Funding

| Funder | Grant reference number | Author |
| --- | --- | --- |
| Zhejiang Natural Science Funds | LR15C060001 | Junling Jia |
| Thousand Young Talents Plan of China | | Liming Wang |
| National Natural Science Foundation of China | 31522026 | Liming Wang |
| Fundamental Research Funds for the Central Universities of China | 2016QN81010 | Liming Wang |

The funders had no role in study design, data collection and interpretation, or the decision to submit the work for publication.

### Author contributions

YY, RH, JY, LW, Conception and design, Acquisition of data, Analysis and interpretation of data, Drafting or revising the article; VZ, Conception and design, Acquisition of data, Analysis and interpretation of data; CW, GC, JJ, Acquisition of data, Analysis and interpretation of data, Contributed unpublished essential data or reagents

### Author ORCIDs

Liming Wang, http://orcid.org/0000-0001-7758-1120

## Additional files

### Supplementary files

• Supplementary file 1. Additional behavioral data and primer sequences. (A) Baseline locomotion for *Figure 1* Summary of our RNAi screen (n = 29–48). For each line, their average daily midline crossing activity when fed *ad libitum* with 5% sucrose ('Baseline locomotion') was listed. The statistical difference between each RNAi line vs. the Control strain was listed. 'Control' is the progeny of the elav-GAL4 driver line crossed to a wild type strain. Data are shown as means ( ± SEM). NS, p>0.05; *p<0.05; **p<0.01; ***p<0.001. (B) Primers for *Figure 6a–b* List of candidate neuropeptide receptor genes, their putative ligands, and the primers used for RT-PCR analysis shown in *Figure 6a–b*. (C). Primers for *Figure 7a–b* List of candidate neurotransmitter related genes and the primers used for RT-PCR analysis shown in *Figure 7a–b*.

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
