## [Decision Letter]

Thank you for submitting your article "Regulation of food-seeking behavior by insulin and glucagon signaling in adult *Drosophila*" for consideration by *eLife*. Your article has been favorably evaluated by K VijayRaghavan (Senior Editor) and three reviewers, one of whom, Mani Ramaswami (Reviewer #1), is a member of our Board of Reviewing Editors.

The reviewers have discussed the reviews with one another and the Reviewing Editor has drafted this decision to help you prepare a revised submission.

Summary:

The authors use simple assays for starvation-induced food seeking and food consumption in *Drosophila* to understand how these behaviors are regulated. They use a quasi-unbiased transgenic RNAi screen to test which of some 31 neuropeptide receptors are required in the nervous system for starvation-induced hyperactivity. Of these they identify one AKHR that has the strongest effect on starvation-induced hyperactivity (though others appear to have significant effects as well). This is significant because AKHR is the receptor for insect glucagon, which is released under starvation conditions. The AKH receptor is expressed in a small number of identified central neurons as well as in sugar sensing labellar neurons: however, it does not appear to be required for starvation induced locomotion in peripheral receptors.

The authors show that:

A) Loss of AKHR or reduced activity of the AKHR positive neurons has the same effect on "food-seeking," without affect food consumption.

B) That the same neurons express Insulin receptors, which have an opposing role on food-seeking. Thus, knocking down insulin signaling in these cells enhances starvation induced food seeking.

C) The same neurons also express octopamine and that octopamine biosynthetic machinery is required in these cells for starvation-induced hyperactivity.

Although some elements of the story were already published (Yang et al., 2015), the current study is a comprehensive synthesis linking hormone modulation to octopamine release. Together, they identify key central sensors of starvation and satiety that control food seeking behaviors and show that these are different from those that control food consumption. It is notable that they engage deeply with mammalian literature, making parallels as appropriate across biological systems.

Major comments:

1) The manuscript should be more judicious in its use of the term "food-seeking." In previous publications, the behaviour is called starvation triggered hyperactivity or locomotion, which is appropriate. Hyperactivity could increase the chance of finding food, without being "food-seeking." The authors could certainly speculate, propose and discuss that this implies increased food-seeking, but without further evidence, the terminology used should not imply that the purpose of the behavior is known or that this is targeted in any way.

2) The authors should verify using either existing genetic mutants or antibody reagents that the central brain neurons identified are indeed AKHR+ as they claim. The RNAi experiments could and should be augmented by use of a genetic mutant and the RT PCR by use of an antibody. Two groups have generated AKHR mutants (Zipursky lab and Kuhnlein lab), UAS-AKHR transgenes, and AKHR antibodies. So there are a number of available reagents that can be used.

3) The work should be complemented by experimental activation (by Chrimson or TRPA1) of the AKHR neurons to determine a direct effect of these neurons on motor activity +/- starvation conditions. Time allowing the authors should also ask whether AKHR+ neurons activation in neurons depleted of octopamine (TBH RNAi or mutants) also causes the flies to increase their motility? The prediction would be that they would not, that the final read-out is octopamine release.

4) The work shows that AKHR is involved not AKH as stated in the last paragraph of the subsection “A neuron-specific RNAi screen identified AKHR required for starvation-induced hyperactivity”. What is the chance that there is a second ligand? However, a previous study by Lee and Park (2004) showed that starvation increases locomotion activity in *Drosophila* and this hyperactivity is absent in flies lacking AKH releasing cells. It should be acknowledged that the AKHR result is consistent with the previous AKH result.

[Editors' note: further revisions were requested prior to acceptance, as described below.]

Thank you for submitting your article "Regulation of starvation-induced hyperactivity by insulin and glucagon signaling in adult *Drosophila*" for consideration by *eLife*. Your article has been reviewed by one peer reviewer, and the evaluation has been overseen by Mani Ramaswami as Reviewing Editor and K VijayRaghavan as the Senior Editor. The reviewers have opted to remain anonymous.

The reviewers have discussed the reviews with one another and the Reviewing Editor has drafted this decision to help you prepare a revised submission.

The results from this study show that AKH and its receptor AKHR are required for starvation-induced hyperactivity. Thus, experimentally, the data are interesting and valuable. However, the main concern from the previous review, that the experimental results need to be interpreted more carefully and critically, has not yet been addressed satisfactorily.

Crucially, the authors overstate the findings as indication that AKH signaling is required for "food-seeking" in *Drosophila*. The language needs to be toned back consistently to better integrate the authors findings with previously published work, and to distinguish between what is actually "shown" by the work and what the authors speculate this may be informing us about Biology.

The authors should include a model summarized below, which all the reviewers agree to be the most parsimonious and probable interpretation of available data.

Upon starvation animals may exhibit two physiological/ behavioral changes, both of which contribute to food-seeking (or foraging or food finding) behavior. One is a change in sensory responses (for example, the change in gustatory neurons as shown by David Anderson and Kristin Scott, and the increased in the sensitivity of Or42b ORNs and the decrease in the sensitivity of Or85a ORNs shown by Jing Wang). Another of which is an increase in locomotion activity as shown in this study and the previous publications. Starvation-induced hyperactivity may simply facilitate food finding (by allowing a wider area to be scanned by the fly) while the targeting aspects are provided by a change in the chemosensory system.

There is no evidence from this study that AKH or its receptor is required for anything beyond the hyperactivity. The fact that starved flies tend to accumulate around food and display a reduced walking speed (shown in Figure 3) merely supports the notion that food (sugar in this experiment) consumption reduces the hyperactivity through the metabolic hormone insulin as shown in the current study that the AKHR-positive neurons also express InR. The findings of the current study are interesting, but they must be presented and discussed more conservatively and in context of what is already known in the field.

Before the paper can be accepted, the authors must do a better job of addressing the following non-exhaustive list of issues focused on this major concern, which was also "major comment #1" in the previous decision letter.

1) Abstract. First sentence "Starvation induces sustained increase in locomotion, which resembles food-seeking behavior and facilitates food localization and acquisition". This should be changed to "Starvation induces a sustained increase in locomotion, which acts to facilitate food localization and acquisition".

2) Introduction. "We have also shown that starvation induced hyperactivity facilitates food localization and acquisition, partly resembling food-seeking behavior in flies". This needs to be revised accordingly (the "partly resembling food-seeking behavior in flies" phrase should be removed).

3) In general, the Introduction should be revised to be more precise and scholarly. It should be framed around the concept that, food seeking behavior has sensory and locomotion aspects. For the sensory aspect, both olfactory and gustatory contributions must be included. The authors should also discuss the previous papers on AKH with the specific goal of discussing what is not known about the starvation-induced AKH-mediated hyperactivity.

4) Title of the section "Neuronal AKHR is required for starvation-induced food seeking" needs to be changed accordingly. The whole section must be edited to deliver a consistent accurate and balanced message.

5) The Discussion section should also be revised in the same manner.

6) Figure 8I shows that sNPF and NPF are only involved in food consumption. This is incorrect. Previous work has shown that both are involved in foraging behavior. The text should acknowledge that food seeking is separated into sensory and locomotion aspects.

---

## [Author Response]

*Major comments:*

*1) The manuscript should be more judicious in its use of the term "food-seeking." In previous publications, the behaviour is called starvation triggered hyperactivity or locomotion, which is appropriate. Hyperactivity could increase the chance of finding food, without being "food-seeking." The authors could certainly speculate, propose and discuss that this implies increased food-seeking, but without further evidence, the terminology used should not imply that the purpose of the behavior is known or that this is targeted in any way.*

Following the suggestions from the reviewers, we have revised the manuscript and toned down our statement. We have now used “starvation-induced hyperactivity” to describe our behavioral assays throughout the main text. We have also changed the title too.

Meanwhile, as suggested by the reviewers, we have discussed the potential similarities between starvation-induced hyperactivity and food-seeking behavior in the manuscript. Briefly, our rationale is starvation-induced hyperactivity is a simple yet reliable measure of food seeking, because: a) upon starvation, hyperactive flies are directed towards food sources, suggesting their hyperactivity is to locate food (Figure 3); and b) starvation-induced hyperactivity can be suppressed by food cues (even non-nutritive, palatable food cues), suggesting that this behavior indeed has a clear goal for food search (Yang et al. PNAS 2015).

*2) The authors should verify using either existing genetic mutants or antibody reagents that the central brain neurons identified are indeed AKHR+ as they claim. The RNAi experiments could and should be augmented by use of a genetic mutant and the RT PCR by use of an antibody. Two groups have generated AKHR mutants (Zipursky lab and Kuhnlein lab), UAS-AKHR transgenes, and AKHR antibodies. So there are a number of available reagents that can be used.*

The reviewers asked us to a) confirm the phenotype of neuronal knock-down of AKHR by using AKHR^-/-^ mutants; b) confirm the brain expression of AKHR gene by antibody staining.

a) We have obtained new data using AKHR^-/-^mutants. As shown in Figure 2, AKHR^-/-^ mutants exhibited no increase in locomotion upon starvation, further confirming that AKHR is required for starvation-induced hyperactivity.

Notably, besides the nervous system, AKHR is also expressed in the fat body of fruit flies. We thus also tested the effect of neuronal vs. fat body AKHR knock-down. Unlike neuronal knock-down (Figure 3), fat body knock-down of AKHR did not block starvation-induced hyperactivity (Figure 3—figure supplement 2). These data provide an additional piece of evidence that neuronal AKHR but not fat body AKHR is required for this behavior.

b) We have also tried to get an AKHR antibody. As the reviewers correctly pointed out, Dr. Larry Zipursky’s lab has generated an AKHR antibody as shown in Bharucha and Zipursky J Exp Biol 2008. Unfortunately, this reagent is no longer available per feedback from the Zipursky lab. Generating and validating a new AKHR antibody will take a significant amount of time and will stop us from resubmitting the manuscript within a reasonable time frame.

As an alternative approach to verify AKHR expression in the brain, we isolated the GFP^+^ cells labeled by our AKHR-split-GAL4 from the fly brain, and developed a protocol to conduct single-cell RNA-seq. Consistent with our RT-PCR results, we found that these GFP^+^ brain cells showed high level AKHR expression (Figure 7). As a control for specificity, we also showed that genes that are specifically expressed in tissues outside the central nervous system (such as Or67d and Gr5a in the peripheral nervous system, Slimfast in the fat body, and MyoD in muscle) are not expressed, or expressed at very low levels, in these GFP^+^ brain cells (Figure 7).

Taken together, although we could not verify AKHR expression by antibody staining, we have three independent lines of evidence to support the brain expression of AKHR: GFP expression under the control of endogenous AKHR promoter (Figure 5); single-cell RT-PCR (Figure 7); and single-cell RNA-seq (Figure 7).

Notably, both our single-cell RT-PCR and RNA-seq experiments identified dInR, the receptor of DILPs, in these GFP^+^ neurons (Figure 7). The expression of dInR in AKHR^+^ neurons is verified by antibody staining (Figure 7—figure supplement 1). This provides an additional validation of our single-cell analysis.

*3) The work should be complemented by experimental activation (by Chrimson or TRPA1) of the AKHR neurons to determine a direct effect of these neurons on motor activity +/- starvation conditions. Time allowing the authors should also ask whether AKHR+ neurons activation in neurons depleted of octopamine (TBH RNAi or mutants) also causes the flies to increase their motility? The prediction would be that they would not, that the final read-out is octopamine release.*

We have added a new figure (Figure 6) to show the neuronal activation results. To summarize, we have found that a) artificial activation of AKHR^+^ neurons accelerated the onset of hyperactivity upon starvation; b) artificial activation of AKHR^+^ neurons did not affect the frequency and duration of feeding, as well as the meal size. Taken together, AKHR^+^ neurons are both necessary and sufficient for starvation-induced hyperactivity, and are not involved in the regulation of food consumption.

We have used the ectopic expression of NaChBac, a low-threshold sodium channel, to perform our neuronal activation experiments. NaChBac is a commonly used tool for neuronal activation in flies (for example, see Dus et al. Neuron 2014; Shang et al. PNAS 2008; Zhou et al. Nature Neuroscience 2008).

Besides NaChBac, we have also tried thermogenic neuronal activation by TRPA1. However we found that acute ambient temperature shift disrupted the locomotor pattern of these flies so that thermogenic control was not an appropriate way for neuronal activation in our assays. Chrimson likely has a similar problem since light stimulus can cause a startle response in free moving flies (Klapotke et al. Nature Methods 2014).

We have shown that eliminating octopamine signaling in AKHR^+^ neurons blocks starvation-induced hyperactivity (Figure 7D-F), so as eliminating AKHR in octopaminergic neurons (Figure 7G). These results confirm that octopamine release from AKHR^+^ neurons is indeed required for AKHR^+^ neurons to induce starvation-induced hyperactivity.

*4) The work shows that AKHR is involved not AKH as stated in the last paragraph of the subsection “A neuron-specific RNAi screen identified AKHR required for starvation-induced hyperactivity”. What is the chance that there is a second ligand? However, a previous study by Lee and Park (2004) showed that starvation increases locomotion activity in Drosophila and this hyperactivity is absent in flies lacking AKH releasing cells. It should be acknowledged that the AKHR result is consistent with the previous AKH result.*

We have acknowledged the paper by Lee and Park in the revised manuscript. As the reviewers pointed out, Lee and Park found that ablating AKH^+^ cells eliminated starvation-induced hyperactivity, which is consistent with our manipulations on AKHR and AKHR^+^ neurons.

Furthermore, we have conducted additional experiments to eliminate AKH gene expression. As shown in Figure 2, knock-down of AKH also blocked starvation-induced hyperactivity, phenocopying the results of neuronal knock-down of AKHR and AKHR^-/-^ mutants. These results further suggest that AKHR mediates the behavioral effect of AKH. Taken together, we have added both experimental evidence and previous literature to support the direct link between AKH and AKHR in starvation-induced hyperactivity.

[Editors' note: further revisions were requested prior to acceptance, as described below.]

*The results from this study show that AKH and its receptor AKHR are required for starvation-induced hyperactivity. Thus, experimentally, the data are interesting and valuable. However, the main concern from the previous review, that the experimental results need to be interpreted more carefully and critically, has not yet been addressed satisfactorily.*

*Crucially, the authors overstate the findings as indication that AKH signaling is required for "food-seeking" in Drosophila. The language needs to be toned back consistently to better integrate the authors findings with previously published work, and to distinguish between what is actually "shown" by the work and what the authors speculate this may be informing us about Biology.*

*The authors should include a model summarized below, which all the reviewers agree to be the most parsimonious and probable interpretation of available data.*

*Upon starvation animals may exhibit two physiological/ behavioral changes, both of which contribute to food-seeking (or foraging or food finding) behavior. One is a change in sensory responses (for example, the change in gustatory neurons as shown by David Anderson and Kristin Scott, and the increased in the sensitivity of Or42b ORNs and the decrease in the sensitivity of Or85a ORNs shown by Jing Wang). Another of which is an increase in locomotion activity as shown in this study and the previous publications. Starvation-induced hyperactivity may simply facilitate food finding (by allowing a wider area to be scanned by the fly) while the targeting aspects are provided by a change in the chemosensory system.*

*There is no evidence from this study that AKH or its receptor is required for anything beyond the hyperactivity. The fact that starved flies tend to accumulate around food and display a reduced walking speed (shown in Figure 3) merely supports the notion that food (sugar in this experiment) consumption reduces the hyperactivity through the metabolic hormone insulin as shown in the current study that the AKHR-positive neurons also express InR. The findings of the current study are interesting, but they must be presented and discussed more conservatively and in context of what is already known in the field.*

As suggested by the reviewers, we have toned down our interpretation and discussions of the behavioral data throughout the manuscript. We have now made it clear that starvation modulates food-seeking behavior in two ways (modulation of food perception and starvation-induced hyperactivity), and that the main focus of the present study is only about starvation-induced hyperactivity. We have largely eliminated the use of “food seeking” in the manuscript. We have also acknowledged the recent findings on food-seeking behavior more thoroughly.

More specifically:

In the Introduction section, we have followed the logic flow suggested by the reviewers: the importance and the regulation of food intake (Paragraph 1); the concept of food seeking, and how it is modulated by starvation in two ways (modulation of the chemosensory systems as well as locomotion) (Paragraph 2-3); the relationship between starvation-induced hyperactivity and food consumption (Paragraph 4); as well as a summary of the motivation and key findings of this present study (Paragraph 5).

Specifically, as suggested by the reviewers, we have summarized the recent findings on the modulation of olfactory and gustatory sensory neurons by starvation from Jing Wang, David Anderson, and Kristin Scott’s labs (Paragraph 2). Meanwhile, we have also discussed the previous reports on AKH and octopamine signaling in starvation-induced hyperactivity (Paragraph 3).

In the Results section, we have revised the interpretation of video-recording assays (Figure 3—figure supplement 3, previously Figure 3). We agree with the reviewers that these data did not really demonstrate that neuronal AKHR is required for food seeking, but rather provides an additional piece of evidence that neuronal AKHR is required for starvation-induced hyperactivity.

In the Discussion section, we have now clearly stated that starvation facilitates food seeking by modulating the perception of food cues and by enhancing the locomotion, and that the main focus of the present study is the latter (Paragraph 1).

Last but not least, we have revised our model to reflect all these changes mentioned above. As shown in Figure 7, starvation induces two separate behavioral responses, food consumption and food seeking, the latter of which can be further divided into the targeting component (perception of food cues) as well as the exploration component (hyperactivity). The focus of this present study, as highlighted in red, is only one of the several behavioral responses to starvation.

*Before the paper can be accepted, the authors must do a better job of addressing the following non-exhaustive list of issues focused on this major concern, which was also "major comment #1" in the previous decision letter.*

*1) Abstract. First sentence "Starvation induces sustained increase in locomotion, which resembles food-seeking behavior and facilitates food localization and acquisition". This should be changed to "Starvation induces a sustained increase in locomotion, which acts to facilitate food localization and acquisition".*

*2) Introduction. "We have also shown that starvation induced hyperactivity facilitates food localization and acquisition, partly resembling food-seeking behavior in flies". This needs to be revised accordingly (the "partly resembling food-seeking behavior in flies" phrase should be removed).*

We have removed such statements throughout the manuscript. As suggested by the reviewers, we have now clearly stated that starvation induced two adaptive changes to facilitate food seeking: starvation-induced hyperactivity and starvation-induced changes in the perception of food cues. The focus of this current study is only around starvation-induced hyperactivity, while the progress on the modulation of chemosensory systems is also discussed in depth.

*3) In general, the Introduction should be revised to be more precise and scholarly. It should be framed around the concept that, food seeking behavior has sensory and locomotion aspects. For the sensory aspect, both olfactory and gustatory contributions must be included. The authors should also discuss the previous papers on AKH with the specific goal of discussing what is not known about the starvation-induced AKH-mediated hyperactivity.*

We have now discussed the previous literature on the regulation of olfactory and gustatory sensory systems by starvation. We have also discussed the previous findings on AKH-producing cells and octopamine in starvation-induced hyperactivity.

*4) Title of the section "Neuronal AKHR is required for starvation-induced food seeking" needs to be changed accordingly. The whole section must be edited to deliver a consistent accurate and balanced message.*

We have removed “starvation-induced food seeking” in that section. Along with the reviewers, we agree that the video-recording experiments (Figure 3—figure supplement 3) provide an additional piece of evidence for the role of neuronal AKHR in starvation-induced hyperactivity.

*5) The Discussion section should also be revised in the same manner.*

We have substantially revised the Discussion section, starting from describing the targeting (perception of food cues) and exploration (hyperactivity) components of food-seeking behavior, followed by the summary of our present study. We have also made it clear that our current study is only about the hyperactivity induced by starvation.

*6) Figure 8I shows that sNPF and NPF are only involved in food consumption. This is incorrect. Previous work has shown that both are involved in foraging behavior. The text should acknowledge that food seeking is separated into sensory and locomotion aspects.*

We have revised the model (now Figure 7) to incorporate these valuable comments. Specifically, we have made it clear that starvation modulates food seeking in two ways: one for targeting (modulated perception of food cues) and the other for exploration (hyperactivity). We have also summarized recent findings on the targeting component of food-seeking behavior from Jing Wang, David Anderson, and Kristin Scott’s labs (including NPF and sNPF).